# Poly(A) tail length regulates PABPC1 expression to tune translation in the heart

**Sandip Chorghade[1†], Joseph Seimetz[1†], Russell Emmons[2], Jing Yang[3], Stefan M Bresson[4], Michael De Lisio[2,5], Gianni Parise[6], Nicholas K Conrad[4], Auinash Kalsotra[1,7]***

[1]Department of Biochemistry, University of Illinois, Illinois, United States; [2]Department of Kinesiology and Community Health, University of Illinois, Illinois, United States; [3]Department of Comparative Biosciences, University of Illinois, Illinois, United States; [4]Department of Microbiology, University of Texas Southwestern Medical Center, Dallas, United States; [5]School of Human Kinetics, University of Ottawa, Ottawa, Canada; [6]Department of Kinesiology, McMaster University, Hamilton, Canada; [7]Carl R. Woese Institute of Genomic Biology, University of Illinois, Illinois, United States

**Abstract** The rate of protein synthesis in the adult heart is one of the lowest in mammalian tissues, but it increases substantially in response to stress and hypertrophic stimuli through largely obscure mechanisms. Here, we demonstrate that regulated expression of cytosolic poly(A)-binding protein 1 (PABPC1) modulates protein synthetic capacity of the mammalian heart. We uncover a poly(A) tail-based regulatory mechanism that dynamically controls PABPC1 protein synthesis in cardiomyocytes and thereby titrates cellular translation in response to developmental and hypertrophic cues. Our findings identify PABPC1 as a direct regulator of cardiac hypertrophy and define a new paradigm of gene regulation in the heart, where controlled changes in poly(A) tail length influence mRNA translation.

*For correspondence: kalsotra@illinois.edu

†These authors contributed equally to this work

**Competing interests:** The authors declare that no competing interests exist.

## Introduction

Cellular growth and function depend heavily on protein synthesis, which is often considered a constitutive activity for a cell. However, it is becoming clear that global protein synthesis rates are not always static, that they vary widely among cell types, and that these differences are necessary for normal tissue development and homeostasis (*Buszczak et al., 2014*). Particularly, the rate of protein synthesis in adult heart is one of the lowest amongst different tissues but increases markedly in response to stress and hypertrophic stimuli (*Garlick et al., 1980*; *Lewis et al., 1984*). The molecular basis for these historical observations, however, is still poorly understood.

Translation initiation is the rate-limiting step in protein synthesis (*Aitken and Lorsch, 2012*; *Hinnebusch et al., 2016*; *Sonenberg and Hinnebusch, 2009*). Interactions between the 5' m7GpppN cap structure, the pre-initiation factors (including eIF4A, eIF4E, and eIF4G), and poly(A)-binding protein C1 (PABPC1) form a stable, looped mRNP complex (*Amrani et al., 2008*; *Gallie, 1991*; *Park et al., 2011*; *Safaee et al., 2012*; *Tarun and Sachs, 1996*; *Wells et al., 1998*) that stimulates translation while safeguarding the mRNA from exonucleases (*Coller et al., 1998*; *Gray et al., 2000*; *Kahvejian et al., 2005*; *Lewis et al., 2017*; *Zekri et al., 2013*). Based on these central roles, PABPC1 is thought to be ubiquitously expressed and serve 'house-keeping' roles in protein synthesis.

**eLife digest** Hundreds of thousands of different proteins are needed to build and maintain the cells in the human body. All proteins are produced when copies of genetic information in the form of molecules of messenger RNA (mRNAs) are translated by the ribosome. The rate at which proteins are made varies widely between different tissues and at different times. In particular, the adult heart has one of the lowest rates of protein production, though this rate can increase markedly during exercise and heart disease. The mechanisms that drive this kind of dynamic change in protein production remain unclear. A better understanding of this process would tell scientists more about how and why cells regulate the translation of mRNAs in general, and might uncover new ways for treating heart disease.

Molecules of mRNA are composed of smaller building blocks called nucleotides. All mature mRNAs in humans have a long stretch at one end that contains the nucleotide adenosine – commonly referred to as A for short – repeated 200 to 300 times. This sequence is called the poly(A) tail, and specific proteins can bind to this tail and determine the final fate of the mRNA, such as how efficiently it is translated. One such poly(A) binding protein, called PABPC1, is known to promote mRNA translation.

Now, Chorghade, Seimetz et al. examine how PABPC1 regulates protein production in mice and human cells. The experiments reveal that, before birth, ample amounts of PABPC1 are found in the heart, but that this protein is undetectable in the hearts of adults. Further experiments showed that the levels of the mRNA for PABPC1 in the heart are the same before birth and in adulthood. So why is the mRNA for PABPC1 translated inefficiently in adult hearts? In general, mRNAs with long tails tend to be translated more efficiently compared to those with short tails, and it turns out that the mRNA for PABPC1 has a substantially shorter poly(A) tail in the adult heart. This tail shortening limits the translation of this specific mRNA, which leads to reduced protein production.

Chorghade, Seimetz et al. also showed that the length of the poly(A) tail on the mRNA and the levels of the PABPC1 protein are restored in adult hearts during a condition known as hypertrophy. This state of hypertrophy can be triggered by exercise or heart disease and is marked by an increase in the size of individual heart cells and enhanced protein production. Finally, Chorghade, Seimetz et al. found that experimentally raising the levels of PABPC1 in adult hearts could, by itself, make the heart cells produce more protein and the heart grow more. Further studies will explore if other mRNAs in the heart also undergo similar changes and whether this is a general mechanism for controlling protein production.

Here, we report that PABPC1 protein expression is post-transcriptionally silenced in adult human and mouse hearts through shortening of its mRNA poly(A) tail, which results in reduced polysome association and translation of *Pabpc1* transcripts. The developmental silencing of PABPC1 is cardiomyocyte-specific and reversible. We show that *Pabpc1* poly(A) tail length and protein expression are restored during adult-onset cardiac hypertrophy stimulated by endurance exercise or heart disease. Furthermore, we demonstrate that PABPC1 re-expression and its interaction with eIF4G are necessary and sufficient to globally stimulate translation and physiologic growth of cardiomyocytes. These findings reveal a novel, poly(A) tail-based regulatory mechanism in the heart that dynamically controls PABPC1 expression and subsequent protein synthesis in response to developmental and hypertrophic signals.

## Results and discussion

The association of eIF4F complex with the 5′ m$^7$G cap structure is stabilized through eIF4G-PABPC1 interactions, which promote ribosomal recruitment and translation initiation (*Amrani et al., 2008*; *Gallie, 1991*; *Safaee et al., 2012*; *Tarun and Sachs, 1996*; *Wells et al., 1998*). We have discovered that PABPC1 protein levels in the adult mouse heart are drastically lower relative to the embryonic day (E)17 stage (*Figure 1A,B*). Parallel examination of *Pabpc1* mRNA abundance unexpectedly showed only a modest decrease after birth (*Figure 1B*). A similarly striking reduction in PABPC1 protein, but not mRNA levels, was observed in adult versus fetal human hearts indicating PABPC1

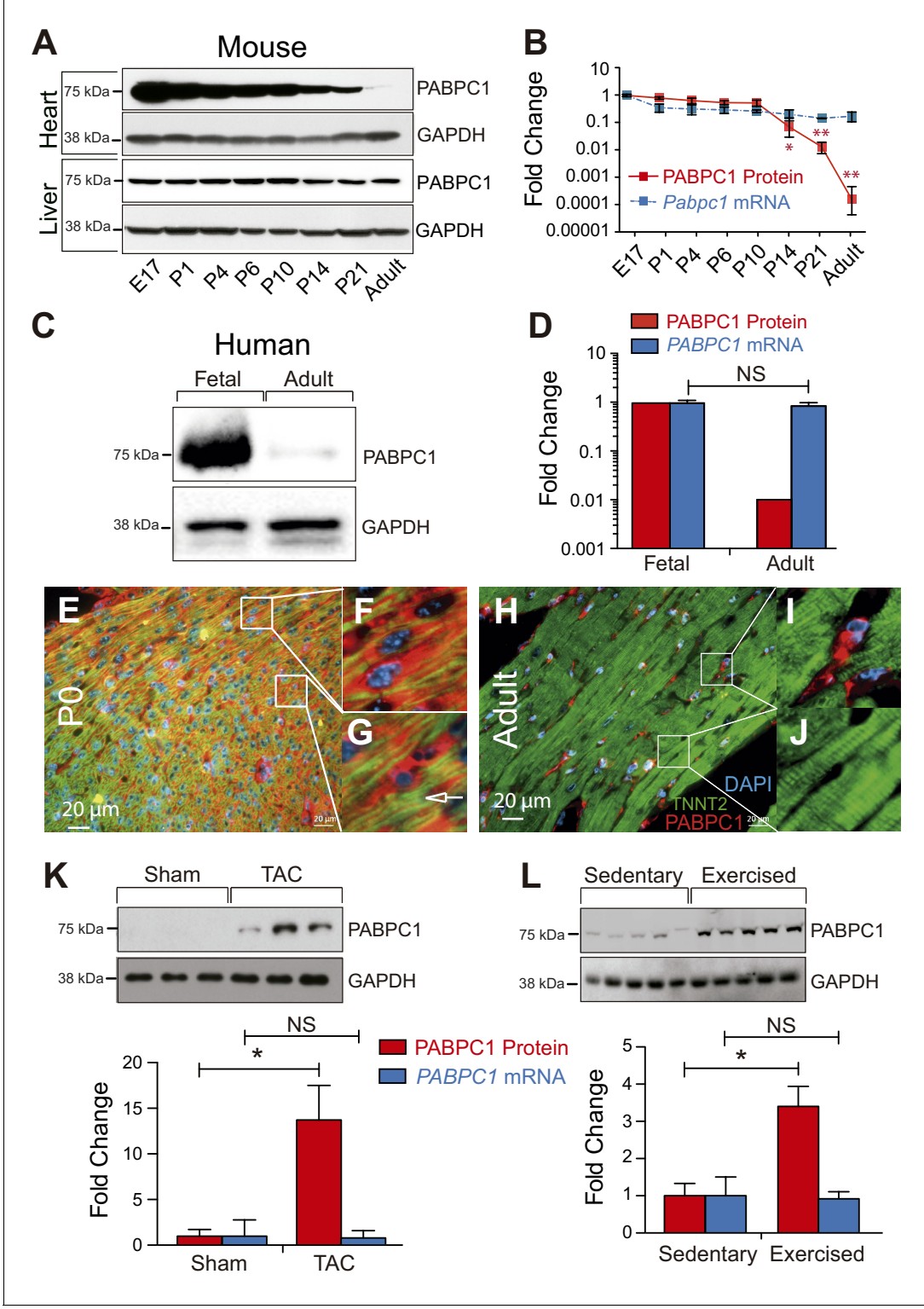

**Figure 1.** PABPC1 is dynamically regulated during cardiac development and hypertrophy. (A–D) Relative quantification of PABPC1 protein (immunoblots) and mRNA (qPCR) levels normalized to GAPDH during mouse heart and liver development (A, B) and in human fetal and adult hearts (C, D). (E–J) Immunofluorescent images of mouse postnatal day 0 (P0) and 8-week-old adult hearts stained for PABPC1 (red), cardiac troponinT (green), and DAPI (blue). Insets G and J show cardiomyocytes, while F and I show non-cardiomyocytes. Immunoblots and quantification of PABPC1 protein and mRNA from wild-type mouse hearts 8 weeks after (K) thoracic aortic constriction (TAC) or (L) 10-week exercise training. Data are mean ± s.d (n = 3); *p<0.05, unpaired two-tailed *t*-test. NS, not significant.

*Figure 1 continued on next page*

*Figure 1 continued*

The following figure supplements are available for figure 1:

**Figure supplement 1.** Post-transcriptional silencing of PABPC1 is muscle-specific.

**Figure supplement 2.** Post-transcriptional silencing of PABPC1 is muscle-specific.

silencing is post-transcriptional and evolutionarily conserved (*Figure 1C,D*). We inspected PABPC1 mRNA and protein abundance in several other mouse and human tissues and determined that the postnatal silencing of PABPC1 is muscle-specific (*Figure 1A* and *Figure 1—figure supplement 1A–D*). Coimmunofluorescent staining of PABPC1 with a cardiomyocyte marker (TNNT2) combined with immunoblot analyses of purified cell types revealed that while PABPC1 is abundantly expressed in all cells within the neonatal heart, it is selectively silenced in adult cardiomyocytes (*Figure 1E,J* and *Figure 1—figure supplements 1E* and *2*).

We sought to determine whether PABPC1 protein is re-expressed in the adult heart during hypertrophy, a condition accompanied by increased size of individual cardiomyocytes, enhanced protein synthesis, and induction of many fetal genes (*Hill and Olson, 2008*; *Maillet et al., 2013*; *Towbin and Bowles, 2002*). We used thoracic aortic constriction (TAC) and endurance exercise training in mice as models of pathologic and physiologic hypertrophy, respectively (*De Lisio and Parise, 2012*; *Kalsotra et al., 2014*). PABPC1 protein levels were markedly induced under both conditions without any change in mRNA abundance (*Figure 1K,L*). These results illustrate that post-transcriptional silencing of PABPC1 in the adult heart is reversed during cardiac hypertrophy.

To probe whether the disappearance of PABPC1 protein in the adult heart is due to a reduction of synthesis, we conducted polysome profiling of E18 and adult mouse hearts (*Figure 2A*). Quantitative PCR (qPCR) analyses of fractionated lysates showed a significant shift of *Pabpc1* mRNAs away from polyribosomes to the mRNP/monosome fractions in the adult hearts, demonstrating reduced accessibility to the translational machinery in contrast to the E18 hearts. Control *Gapdh* mRNA remained associated with polyribosomes at both developmental stages (*Figure 2B*).

Intriguingly, in comparison to E18, the adult heart polysome profiles exhibited a decrease in the number of polyribosomes with a corresponding increase of 80S monosomes, which reflects reduced translation efficiency (*Figure 2A*). To further quantify global protein synthesis rates in vivo, we pulse-labeled wild-type neonates and adult mice with puromycin followed by immunoblotting using an anti-puromycin antibody (i.e. SUnSET assays) (*Goodman et al., 2011*; *Schmidt et al., 2009*). We detected a robust decrease in puromycin-labeled peptides in adult versus neonatal hearts but not in liver tissues (*Figure 2C*). Lower protein synthetic capacity of adult striated muscle compared to other tissues was first observed more than three decades ago (*Garlick et al., 1980*; *Lewis et al., 1984*). Our data showing muscle-specific silencing of PABPC1 thus offers a plausible molecular basis for these historical observations. Low-level protein synthesis in the absence of PABPC1 could presumably result from alternative mRNA circularization mechanisms (*Bukhari et al., 2016*; *Lin et al., 2016*; *Wang et al., 2015*) or translation of some mRNAs in a closed loop-independent manner (*Archer et al., 2015*; *Costello et al., 2015*).

Next, we investigated the molecular mechanisms responsible for inefficient translation of PABPC1. It appears that suppressed PABPC1 translation in the adult heart is microRNA-independent, as we saw no change in PABPC1 protein or mRNA in tamoxifen-inducible, heart-specific adult *dicer* knockouts (*Figure 2—figure supplement 1A,B*), which are defective in microRNA processing (*Kalsotra et al., 2010*). We analyzed available RNA-sequencing data from neonatal and adult mouse cardiomyocytes and fibroblasts (*Giudice et al., 2014*) and found no evidence for alternative splicing or a developmental change in *Pabpc1* 5'- or 3'- untranslated regions (UTRs) in either cell types (*Figure 2—figure supplement 1C*). We reasoned that *trans*-acting factor(s) might bind to the sequence elements within *Pabpc1* UTRs to suppress its translation. To test this hypothesis, we constructed luciferase reporters fused to 5'-, 3'- or both mouse *Pabpc1* UTRs and transfected them into C2C12 myoblasts. C2C12 cells exhibit marked downregulation of endogenous PABPC1 protein (~10 fold) but not mRNA levels when differentiated into myotubes (*Figure 2—figure supplement 2A,B*).

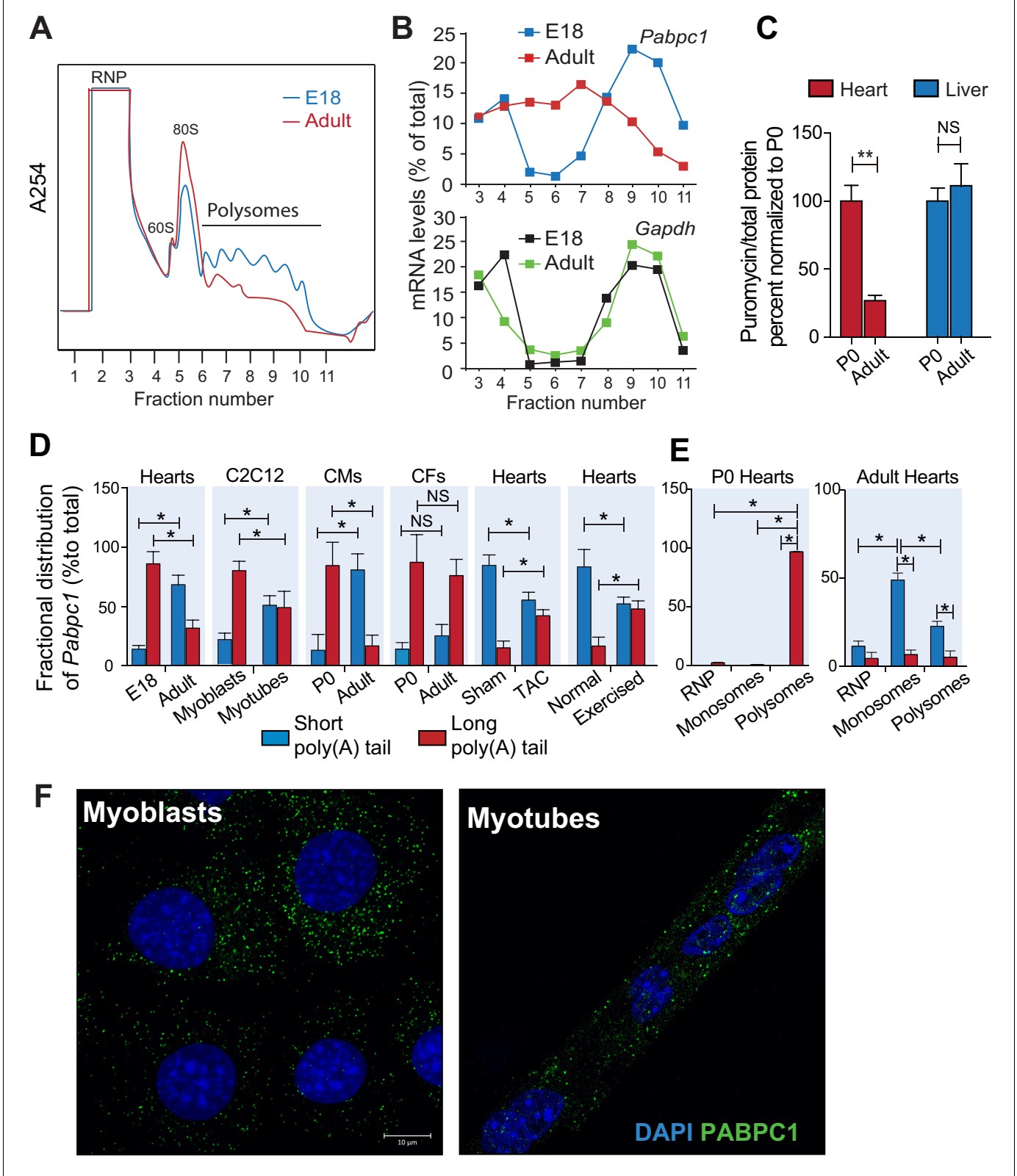

**Figure 2.** Poly(A) tail length determines cell-type and developmental stage-specific translation of PABPC1. (**A**). Polysome profile of embryonic day 18 (E18) and adult mouse hearts. (**B**) Percentage of *Pabpc1* and *Gapdh* mRNAs measured by qPCR in each fraction collected from the polysome profiling. (**C**) Neonatal and 8-week-old adult wild-type mice were pulsed with puromycin through an intraperitoneal injection. Forty-five minutes following

*Figure 2 continued on next page*

*Figure 2 continued*

injection, heart and liver tissues were harvested for immunoblotting with anti-puromycin antibody. De novo protein synthesis was quantified as the ratio of puromycin labeled peptides to total protein. (D) Fractional distribution of *Pabpc1* mRNAs with short and long poly(A) tails in whole heart, C2C12 cells, cardiomyocytes (CMs), cardiac fibroblasts (CFs), whole heart after TAC surgery, and whole heart after exercise (measured by qPCR following poly (A) tail fractionation). (E) Poly(A) tail length status of *Pabpc1* mRNA within P0 and adult heart RNP, monosome, and polysome fractions from sucrose gradients. (F) *Pabpc1* single-molecule RNA-FISH in C2C12 myoblasts and myotubes. Data are mean ± s.d (n = 3); *p<0.05, **p<0.005 unpaired two-tailed *t*-test; NS, not significant.

The following figure supplements are available for figure 2:

**Figure supplement 1.** Regulation of PABPC1 expression in adult heart is independent of miRNAs or alternative splicing.

**Figure supplement 2.** Limited influence for 5' and 3' untranslated regions (UTRs) of *Pabpc1* on luciferase protein translation during C2C12 differentiation.

**Figure supplement 3.** Experimental design of northern blot and RNA isolation based on poly(A) tail length through gradient purification.

Amongst the different reporters, only *Pabpc1* 5'-UTR showed a modest (<2-fold) decrease in luciferase (*Rluc/Fluc*) activity during myoblast-to-myotube differentiation (*Figure 2—figure supplement 2C,D*). Although PABPC1 binding to an A-rich element within its 5'-UTR could auto-regulate its translation (*de Melo Neto et al., 1995*; *Kini et al., 2016*), the relatively mild effects of 5'-UTR in reporter assays along with decreasing PABPC1 protein levels during muscle development argue against auto-regulatory feedback as the primary mechanism inhibiting *Pabpc1* translation.

Besides UTRs, poly(A) tail length is another major determinant for protein synthesis; mRNAs containing longer tails being more stable and translated more efficiently (*Eichhorn et al., 2016*; *Lim et al., 2016*; *Weill et al., 2012*). Therefore, we tested whether *Pabpc1* mRNAs are deadenylated in the adult heart. Both RNaseH cleavage assay and qPCR of long- and short-tailed mRNAs (fractionated by affinity chromatography) demonstrated significant *Pabpc1* deadenylation in adult versus E18 hearts and in myotubes versus myoblasts (*Figure 2C* and *Figure 2—figure supplement 3A–C*). *Pabpc1* poly(A) tail length in E18 hearts was estimated to be ~150 nucleotides (nts), whereas it was reduced to ~20 nts in adult hearts (*Figure 2—figure supplement 3B*). *Gapdh* poly(A) tail length was measured as a control and showed no difference between E18 and adult hearts. Importantly, shortening of *Pabpc1* poly(A) tail in the adult heart was cardiomyocyte-specific; and partly reversed after TAC or endurance exercise (*Figure 2C*). In addition, *Pabpc1* poly(A) tail length was strongly correlated to its association with monosomes and polysomes. At P0, *Pabpc1* mRNAs were almost exclusively long-tailed and in polysomes, whereas adult *Pabpc1* fractionated majorly into short-tailed and monosome fractions (*Figure 2E*). *Gapdh* poly(A) tail length remained unchanged and primarily associated with polysomes at both developmental stages (*Figure 2—figure supplement 3D*). We further explored if inhibition of *Pabpc1* translation could be due to nuclear retention of *Pabpc1* transcripts. Single-molecule RNA FISH, however, showed primarily cytoplasmic staining without any noticeable difference in *Pabpc1* mRNA localization between myoblasts and myotubes indicating poly(A) tail status does not impact *Pabpc1* nucleo-cytoplasmic export in these cells (*Figure 2F*). Together, these results provide compelling evidence that *Pabpc1* poly(A) tail length is dynamically regulated during cardiac development and hypertrophy, and that poly(A) tail shortening limits *Pabpc1* mRNA translation in the adult heart.

Because PABPC1 is upregulated in hypertrophy, we tested whether it is required for stimulus-induced growth of cardiomyocytes. PABPC1-depleted neonatal mouse cardiomyocytes were viable but resistant to isoproterenol (Iso) or triiodothyronine (T3)-induced hypertrophy (*Figure 3A–I*). Metabolic labeling of cardiomyocytes with an alkyne-modified glycine analog, L-homopropargylglycine (HPG), to measure newly synthesized proteins revealed that PABPC1 knockdown inhibited the normal surge in protein synthesis rate evoked by Iso or T3 stimulation (*Figure 3J,K*). Furthermore, we found that PABPC1 deficiency blocked protein but not mRNA upregulation of the hypertrophic markers *Acta1*, *Myh7* and *Anp* (*Figure 3L,M*) indicating that while new protein synthesis is impaired, the transcriptional response to hypertrophic stimuli is still functional in these cells (*Kim et al., 2008*;

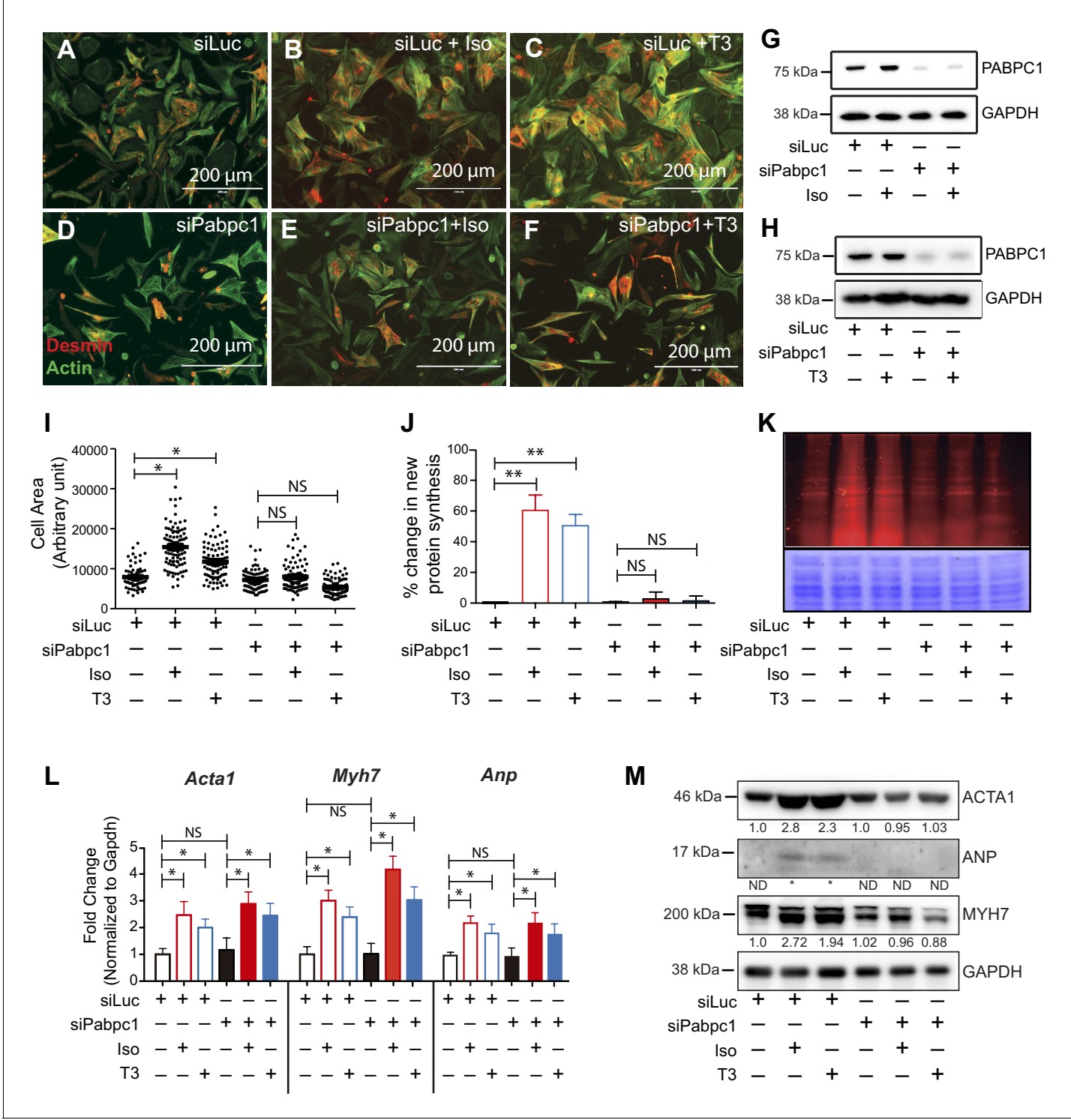

**Figure 3.** Knockdown of PABPC1 in neonatal mouse cardiomyocytes prevents stimulus-induced hypertrophy and protein synthesis. (A–F) Primary cardiomyocytes isolated from newborn mice were transfected with siRNA against control *Luciferase* or *Pabpc1*. Twelve hours following transfection, cells were treated with isoproterenol (Iso) or triiodothyronine (T3) for 36 hr to induce hypertrophy. Cells were stained for Desmin and Actin by immunofluorescence to verify cardiomyocyte identity and measure cell area. (G, H) Immunoblots demonstrating efficient PABPC1 knockdown 48 hr after siRNA treatments with either Iso or T3. (I) Quantification of cell area 36 hr post Iso or T3 treatments. (J, K) Measurement of new protein synthesis using Click-iT homopropargylglycine assay after 2 hr of Iso or T3 treatments. (L) Quantification of mRNA (qPCR) from neonatal cardiomyocytes for each condition shows significant upregulation of mRNA for *Acta1*, *Myh7*, and *Anp* in response to Iso or T3 treatments. (M) Representative immunoblot

*Figure 3 continued on next page*

*Figure 3 continued*

showing that protein levels of ACTA1, MYH7, and ANP are increased after Iso or T3 treatments in the control *Luciferase* knockdown but synthesis is prevented when *Pabpc1* is knocked down. Data are mean ± s.d (n = 3); *p<0.05, **p<0.01, one-way analysis of variance (ANOVA) plus Dunnett's post-hoc test. NS, not significant.

The following source data is available for figure 3:

**Source data 1.** Source data for cell area of cultured neonatal cardiomyocytes treated with siRNA and either Iso or T3.

---

*Sergeeva and Christoffels, 2013*). These results demonstrate that PABPC1 depletion does not affect the transcriptional regulatory circuits or mRNA stability of these hypertrophy markers.

PABPC1 contains four RNA recognition motifs (RRM1-4), a linker region and a C-terminal MLLE domain (*Gray et al., 2000*). While all four RRMs are capable of binding to poly(A) RNA, RRM2 preferentially interacts with eIF4G to stimulate cap-dependent translation (*Safaee et al., 2012*). Hence, we investigated if PABPC1 interactions with eIF4G are necessary for triggering cardiomyocyte hypertrophy and new protein synthesis. The M161A mutation was previously shown to disrupt PABPC1 interactions with eIF4G (*Kahvejian et al., 2005*). However, we decided to use the recently available structural information (*Safaee et al., 2012*) to engineer additional mutations within the RRM2 domain of PABPC1 to completely abolish its interactions with eIF4G (*Figure 4—figure supplement 1A*). We confirmed that although PABPC1$_{mRRM2}$ fails to interact with eIF4G1 in coimmunoprecipitation and in vitro binding assays, it binds to poly(A) RNA with similar affinities as wild-type PABPC1 (*Figure 4—figure supplement 1B–F*).

Next, we carried out rescue experiments wherein endogenous PABPC1 in cardiomyocytes was silenced using a 3'-UTR-targeted siRNA followed by adenoviral transduction of GFP, siRNA-resistant wild type, or PABPC1$_{mRRM2}$ cDNAs (*Figure 4—figure supplement 1G*). Supplementing PABPC1-deficient cardiomyocytes with wild type PABPC1 re-sensitized the hypertrophic growth response to Iso (*Figure 4A–M*), reversed the block in new protein synthesis (*Figure 4N–P*) and restored translation of hypertrophic markers in these cells (*Figure 4Q*). PABPC1$_{mRRM2}$, however, failed to rescue any of these phenotypes (*Figure 4A–Q*) underscoring that PABPC1–eIF4G interactions are essential to evoke cardiomyocyte hypertrophy and stimulate translation in response to hypertrophic signals.

To further determine if PABPC1 upregulation is sufficient to drive cardiac hypertrophy, we generated a doxycycline-inducible, cardiomyocyte-specific PABPC1 transgenic mouse model (*Figure 5* and *Figure 5—figure supplement 1A,B*). Ectopic expression of FLAG-tagged PABPC1 protein in adult mouse hearts showed the expected cytoplasmic distribution (*Figure 5E,F*) and was estimated approximately 12-fold higher over endogenous levels (*Figure 5—figure supplement 1C,D*). Remarkably, forced PABPC1 expression in the adult myocardium led to increased atrial and ventricular size (*Figure 5A–D*), a significant elevation in heart-to-body weight ratios, and larger cardiomyocyte cross-sectional areas compared to uninduced littermate controls (*Figure 5G–J*). We also observed significant increase in global protein synthesis rates along with upregulation of many physiological but not pathological hypertrophic markers in PABPC1 transgenic hearts (*Figure 5K,L*). Notably, PABPC1 transgenic mice did not exhibit premature lethality or cardiac dilation even when induced for 6 months. Moreover, long-term PABPC1 induction did not cause a drop in performance in treadmill tests, deterioration in cardiac contractility, or deficits in systolic and diastolic function (*Figure 5—figure supplement 1E,F*). Also, histologically, we did not observe any myofiber disarray or fibrosis (*Figure 5* and *Figure 5—figure supplement 2*) suggesting hypertrophy in these animals is compensated and does not progress to heart failure, but rather mimics the physiologic form.

In conclusion, we have uncovered that poly(A) tail length shortening suppresses *Pabpc1* mRNA translation in mature cardiomyocytes, thereby reducing overall protein synthesis rates in the adult heart. Intriguingly, despite having a short poly(A) tail, *Pabpc1* transcripts remain stable and are not completely deadenylated or degraded. This might be due to binding of *trans*-acting factor(s), the presence of RNA structural element(s), or RNA modifications that act to stabilize *Pabpc1* in adult cardiomyocytes (*Lewis et al., 2017*). We further demonstrate that PABPC1 protein expression and poly(A) tail length are partially restored during adult-onset cardiac hypertrophy to activate new protein synthesis and physiologic growth of cardiomyocytes. PABPC1 was also shown to be both necessary and sufficient to drive cardiomyocyte hypertrophy. Thus, dynamic control of PABPC1 serves an

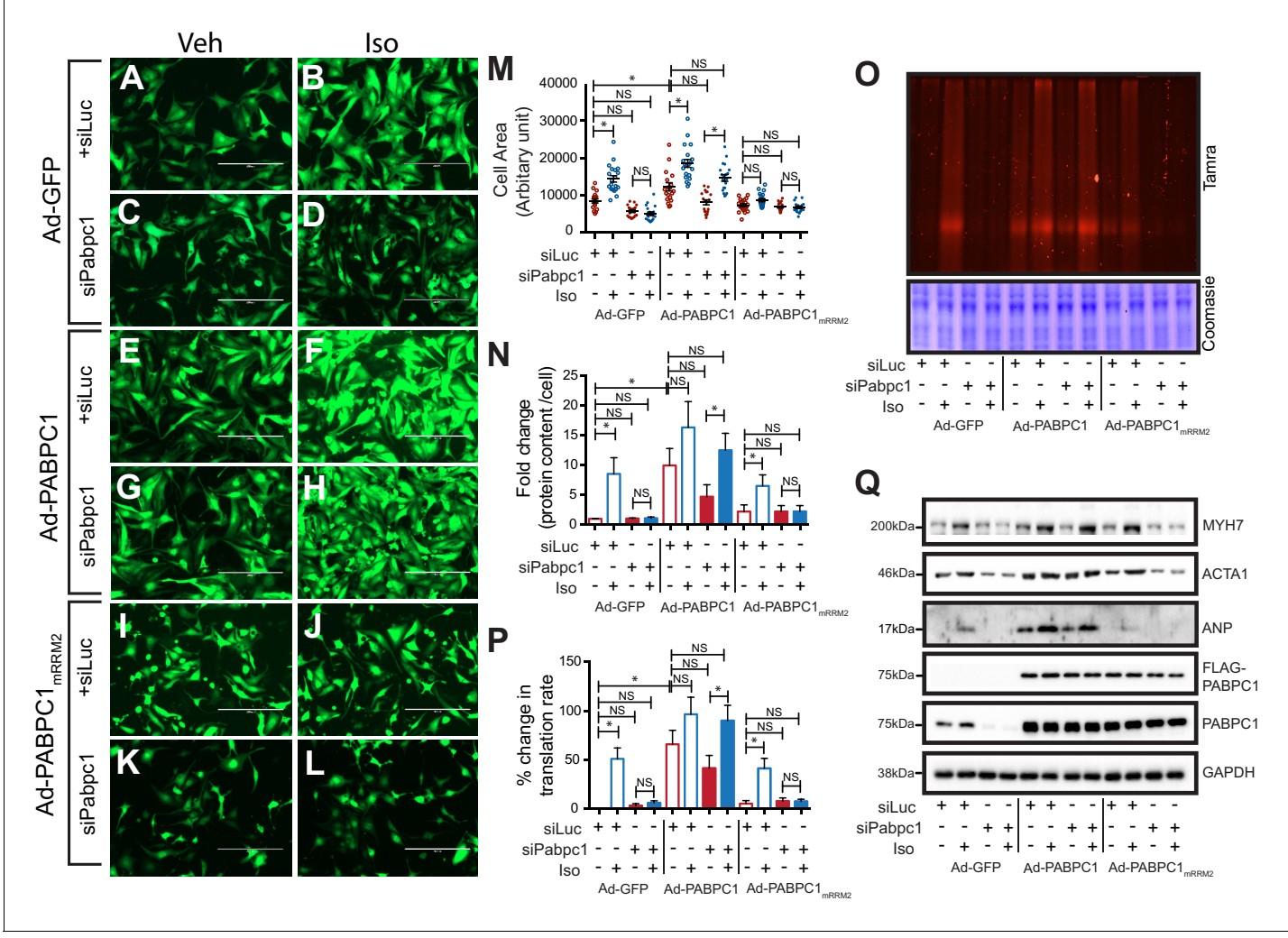

**Figure 4.** PABPC1–eIF4G1 interactions control stimulus-induced new protein synthesis and hypertrophy. (A–L) Representative images of neonatal cardiomyocytes infected with adenovirus expressing GFP, wildtype PABPC1, or a PABPC1 RRM2 mutant (that does not interact with eIF4G1), transfected with siRNA against endogenous *Pabpc1* or *Luciferase*, and treated with isoproterenol (Iso) or vehicle (Veh). Quantification of (M) cell areas, (N) total protein content, (O, P) rate of new protein synthesis measured by Click-iT homopropargylglycine fluorescence assay after respective treatments. (Q) Representative immunoblots of hypertrophy markers. Data are mean ± s.d (n = 3); *p<0.05, one-way analysis of variance (ANOVA) plus Dunnett's post-hoc test. NS, not significant.

The following source data and figure supplement are available for figure 4:

**Source data 1.** Source data for cell area of cultured neonatal cardiomyocytes after treatment with siRNA, adenovirus, and Iso.

**Figure supplement 1.** PABPC1mRRM2 can bind to poly(A) RNA but does not interact with eIF4G1.

adaptive role in stimulating a beneficial form of cardiac hypertrophy that may be physiologically advantageous to the failing or dilated myocardium.

Two genome-wide studies found a weak correlation between poly(A) tail lengths and translational efficiency of mRNAs in cell culture (*Chang et al., 2014*; *Subtelny et al., 2014*). While Subtelny et al. observed clear coupling between poly(A) tail length and translation at early developmental stages in *Xenopus* and zebra fish, the association became less apparent in the adults. Thus, it is plausible that at steady states, poly(A) tail length is less critical for translation but becomes a determinant in certain tissue contexts, developmental stages, cell cycle regulation, daily rhythmic oscillations of protein synthesis, or cellular stress (*Besse and Ephrussi, 2008*; *Kojima et al., 2012*; *Park et al., 2016*).

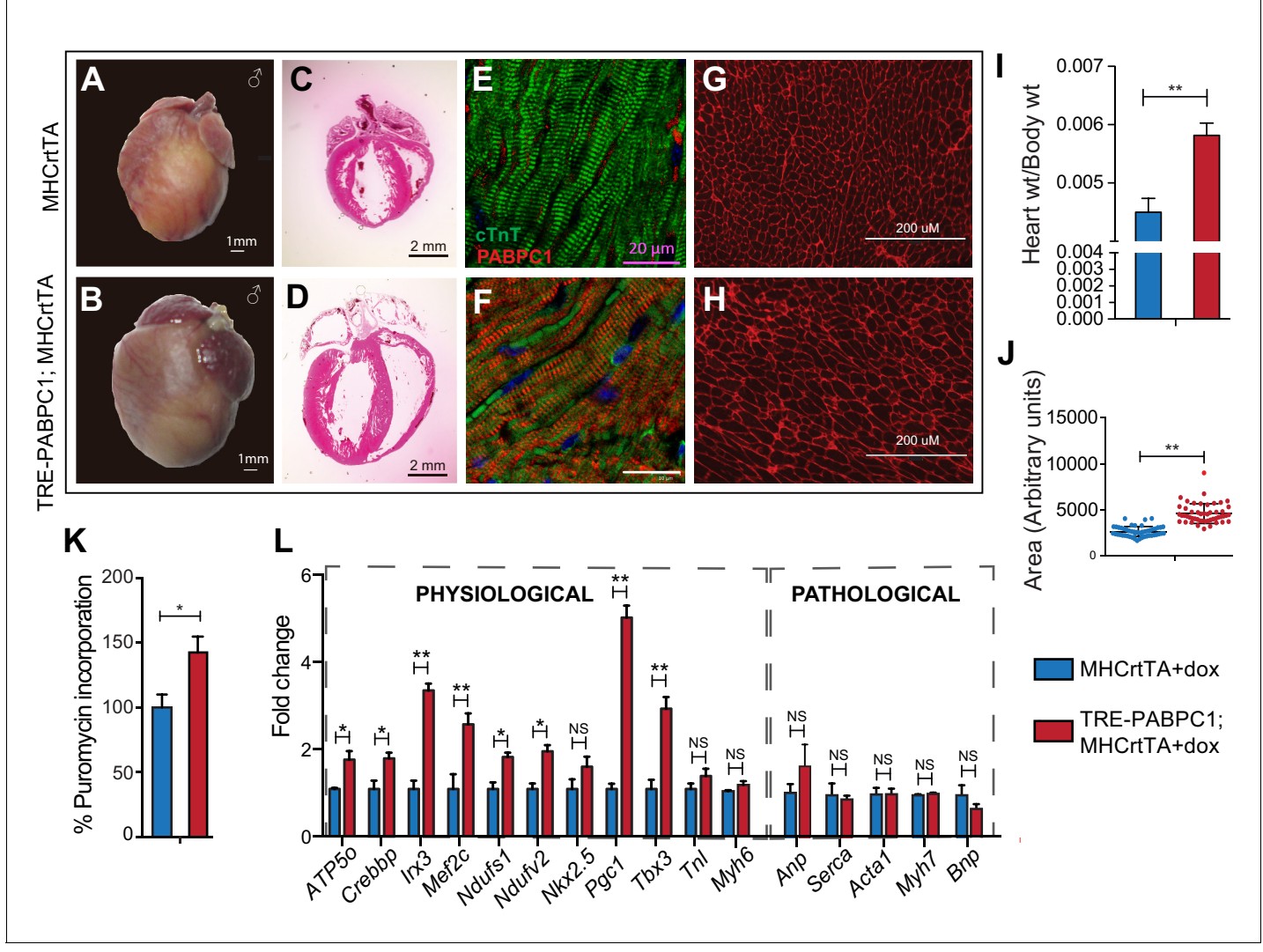

**Figure 5.** Forced expression of PABPC1 in adult cardiomyocytes induces physiologic hypertrophy. (**A–H**) Representative whole heart, H&E, immunofluorescent, and WGA-stained sections of 2-week doxycycline (Dox)-induced MHCrtTA transgenic controls and TRE-PABPC1; MHCrtTA bitransgenic mice. (**I**) Heart-to-body weight ratios (n = 6). (**J**) Cell area quantified from WGA-stained sections (n = 3). (**K**) Global rate of translation based on puromycin incorporation in hearts of injected mice (n = 6). (**L**) Relative mRNA levels of indicated physiological and pathological hypertrophy markers normalized to GAPDH (qPCR, n = 9). Data are mean ± s.d; *p<0.05 unpaired two-tailed *t*-test. NS, not significant.

The following source data and figure supplements are available for figure 5:

**Source data 1.** Source data for cardiomyocyte areas performed on WGA stained heart tissue sections of 2-week doxycycline-induced MHCrtTA transgenic controls and TRE-PABPC1; MHCrtTA bitransgenic mice.

**Source data 2.** Source data for *Figure 5—figure supplement 1*.

**Figure supplement 1.** Generation of tetracycline-inducible, heart-specific PABPC1 transgenic mouse model.

**Figure supplement 2.** Induced expression of cardiomyocyte specific PABPC1 does not lead to damage.

---

Furthermore, stronger relationships between poly(A) tail length and mRNA translatability may emerge under conditions where PABPC1 protein concentrations are limiting; our data from adult cardiomyocytes supports this hypothesis.

Taken together, our findings not only provide insight into the long-standing question of how the heart regulates protein synthesis during development and hypertrophy but also provide a new direction to explore therapeutic interventions. These results set the stage for elucidating the signaling cascades that regulate *Pabpc1* poly(A) tail length in cardiomyocytes, determining mechanistically how those dynamics are achieved, and probing the general roles for poly(A) tail length in translation control within the heart and other somatic tissues.

## Materials and methods

### Animal models and human samples

Mouse PABPC1 cDNA containing an N-terminal FLAG-tag was expressed from a transgene with a TRE/minimal CMV promoter, a genomic fragment including α -MHC untranslated exons 2 and 3 with intron 2 (*Kistner et al., 1996*), the *Pabpc1* ORF and the bovine growth hormone polyadenylation site and 3′ flanking genomic segment for proper mRNA 3′ end formation. The linearized transgene construct was subjected to pronuclear injection using standard methods to generate PABPC1 transgenic mice that were maintained on an FVB background. MHC-rtTA transgenic mice (FVB/N-Tg (*Myh6*-rtTA)1Jam) expressing a codon-optimized rtTA variant specifically in heart were commercially obtained (RRID: MMRRC_010478) (*Valencik and McDonald, 2001*). All mice reported were the F1 progeny of TRE-PABPC1 and MHC-rtTA matings and were, therefore, hemizygous for one or both transgenes. PABPC1 expression in 8- to 12-week-old bitransgenic animals was induced through doxycycline (Dox) in the food (2g Dox/kg food, Harlan, KY). Cardiac-specific *Dicer* knockouts were generated by crossing *Dicer*$^{f/f}$ mice, with Tam-inducible MerCreMer transgenic mice as previously reported (*Kalsotra et al., 2010*). DNA was extracted from tail clips using DirectPCR lysis reagent (Viagen Biotech, Los Angeles, CA) and genotyped by PCR using transgene-specific primers (*Supplementary file 1*). Both male and female mice and littermate controls were used whenever possible. Human fetal (22-week old) and adult (51-year-old Caucasian male) heart RNAs and proteins were purchased from Clonetech Laboratories, Inc., Mountain View, CA We followed the NIH guidelines for use and care of laboratory animals, and all experimental protocols were approved by IACUC (Institutional Animal Care and Use Committee at University of Illinois, Urbana-Champaign.

### Maximal aerobic exercise test to estimate VO$_{2max}$

Mice performed an exhaustive acute exercise bout on a motorized treadmill (Columbus Instruments, Columbus, OH) to determine exercise capacity in Dox-induced TRE-PABPC1 $\times$ MHC-rtTA (n = 15) and control MHCrtTA (n = 13) mice using a previously validated equation (*Fernando et al., 1993*). Prior to the maximal exercise test, mice were acclimated to treadmill running for 2 concurrent days with the following protocol: warm-up 5 min at 8 m/min, 5 min exercise at 10 m/min, and 5 min cool-down at 8 m/min at 0% incline. Three days later, mice performed an exhaustive, incremental exercise bout that began at 11 m/min and increased 1 m/min every 2 min until exhaustion as previously described (*De Lisio and Parise, 2012*). VO$_2$ was determined for each stage using the following equation:

$$Predicted\ VO_2(ml/min) = 0.127 \times weight(g) + (0.040 \cdot running\ speed(m/min)) - 0.974$$

Exhaustion was determined if mice failed to continue running after spending 5 s off of the treadmill bout and not responding to manual stimulation. The researcher evaluating exhaustion was blinded to mouse genotype.

### Exercise training protocol

PABPC1 expression was determined in the hearts of exercise trained C57BL6/J from a previous study (*De Lisio and Parise, 2012*). Six-week-old C57BL6/J mice were randomized into sedentary (SED, n = 12) or endurance trained (EX, n = 10) groups and performed a progressive exercise training program on a motorized treadmill (Columbus Instruments, Columbus, OH). Mice were trained 3 days/ week (M/W/F) for 8 weeks. The exercise protocol consisted of: 10-min warm-up at 12 m/min, 45-min training period that began at 14 m/min (week 1) that increased to a speed of 22 m/min (week 8), and 5-min cool-down at 10 m/min. Mice were trained at the same time each day and sedentary mice were placed onto the treadmill to mimic the stress of handling and treadmill exposure.

## Cardiac function tests

Transthoracic echocardiography was performed on lightly anesthetized mice using 1.5% isoflurane mixed with 95% oxygen as previously reported (*Kalsotra et al., 2010*). Mice were stabilized on a heated platform and taped to ECG electrodes. Evaluation of cardiac function was done using a Visual Sonics Vivo 770 ultra sound using a 30 MHz probe. Two-dimensional guided M-mode tracings were recorded in both parasternal long and short axis views at the level of papillary muscles. Image analysis was done using Visual Sonics software version 2.3.0.

## Histology and immunohistochemistry

Heart tissues from doxycycline-induced MHCrtTA (*n* = 3) transgenic and TRE-PABPC1; MHCrtTA bitransgenic mice (*n* = 3) were harvested and fixed overnight in 10% neutral-buffered formalin, embedded in paraffin, and sectioned (3 µm thickness). Hematoxylin and eosin (H&E) and trichrome stainings were performed using standard histological methods as previously described (*Bhate et al., 2015*). For immunohistochemistry, unstained slides were deparaffinized in xylene (two treatments, 5 min each), rehydrated sequentially in ethanol (2 min each in 100%, 95%, and 80%), and washed for 3 min in water. Antigen retrieval was performed by boiling the sections in Tris buffered solution (20 mM Tris-Cl pH 9, 1 mM EDTA, 0.02% Tween 20) for 20 min at 111°C in a steam cooker then cooled for 20 min under tap water. For ki-67 staining, the endogenous peroxidase activity was quenched with a solution of 3% hydrogen peroxide solution (Fisher Scientific). After washing, sections were blocked (2% normal goat serum, 1% bovine serum albumin (BSA), 0.1% Triton X-100, 0.05%Tween 20 in 1X ~ PBS) for 30 min and incubated with primary anti-Ki-67 antibody at 4°C for 12 hr. After several washes, sections were incubated with HRP-conjugated goat anti-mouse IgG light-chain-specific antibody for 2 hr. For visualization of signal, DAB kit (Vector Labs) was used according to the manufacturer's instructions. All intermediate washing steps were done using 1X PBS, 0.5% Tween 20, pH 7.2 (1X PBST), and all antibodies were diluted in 1X PBST with 1% bovine serum albumin. Slides were sealed with a coverslip after lightly counterstaining with hematoxylin and photographed with an EVOS XL microscope. Wheat germ agglutinin stain (WGA, *L4895 SIGMA*) was used to stain heart cross-section according to manufacturer's instructions.

## In vivo SUnSET assay

Proteins were labeled using a protocol adapted from the SUnSET method (*Goodman et al., 2011*; *Schmidt et al., 2009*). P0 and Adult mice were injected with puromycin made in sterile PBS (0.04 µmol/ per g of body weight). After 45 min hearts and liver were harvested and protein lysates were prepared. Proteins were separated by 10% SDS-PAGE. Puromycin-labeled peptides were identified using the mouse monoclonal antibody 12D10 (1: 5000 dilution). Protein synthesis levels were determined by densitometry analysis of whole lanes. Normalization of the free puromycin was analyzed as previously described (*Goodman et al., 2011*).

## Polysome gradient fractionation

Whole hearts (pooled batches of 2 to 3 hearts) were extracted and washed in ice-cold PBS containing cyclohexamide. Blood was removed by squeezing the heart with blunt forceps and quickly pulverized under liquid nitrogen using previously cooled, RNAse free mortar and pestle. The powder obtained was transferred to a 10 cm plate, previously cooled on dry ice for 10 min. Afterward, 1 mL of lysis buffer (10 mM Tris-HCl at pH 8.0, 150 mM NaCl, 5 mM MgCl2, 1% Nonidet-P40, 40 mM dithiothreitol, 500 U/mL RNAsin [Promega], 40 mM VRC [New England Bio Labs]) supplemented with 1% deoxycholate [Fluka] was added to the tissue powder. Next, re-suspended powder was scraped from the plate and transferred to a 2-mL tube with pipetting 10x to lyse the cells. The cell nuclei were removed by centrifugation (12,000 *g*, 10 s, at 4°C), and the supernatant was supplemented with 500 µL of 2X extraction buffer (0.2 M Tris-HCl at pH 7.5, 0.3 M NaCl), 150 µg/mL cycloheximide, 650 µg/mL heparin, and 10 mM phenyl-methylsulfonyl fluoride, and centrifuged (12,000 *g*, 5 min, at 4°C) to remove mitochondria and membranous debris. The supernatant was layered onto a 10 mL linear sucrose gradient (15%–45% sucrose [w/v], supplemented with 10 mM Tris-HCl at pH 7.5, 140 mM NaCl, 1.5 mM MgCl2, 10 mM dithiothreitol, 100 µg/mL cycloheximide, 0.5 mg/mL heparin) and centrifuged in a SW41Ti rotor (Beckman) for 120 min at 38,000 rpm and 4°C, with the brake off. Polysome profiles were recorded using a UA-6 absorbance (ISCO) detector at 254 nm. Fractions

(12 × 1 mL) were collected and RNAs were recovered by extraction with an equal volume of Trizol. RNAs were reverse transcribed using random hexamer primers and Maxima Reverse Transcriptase kit (Thermo Scientific). The cDNA was diluted to 25 ng/µL with nuclease free water and used for gene-specific qPCR assays.

## Protein isolation and western blot analysis

Proteins from hearts or purified cell fractions were isolated as previously described (*Bhate et al., 2015*). In brief, frozen heart tissue or purified cell fractions were homogenized with cold homogenization buffer (10 mM HEPES-KOH, pH 7.5, 0.32 M Sucrose, 5 µM MG132, 5 mM EDTA-free Proteinase inhibitor [Pierce 88666, Thermo Fisher]) using a bullet blender (Next Advance). Samples were sonicated in the presence of 0.1% SDS and clarified by centrifugation (20,000 rcf at 4°C). The protein content was measured using the BCA protein assay kit (Thermo Scientific). Protein lysates (100–150 µg of protein loaded per lane) were resolved by 10% SDS– polyacrylamide gel electrophoresis gels and transferred onto PVDF membranes (Immobilon, Millipore). Membranes were blocked in Tris-buffered saline (TBS) containing 5% non-fat dry milk and 0.2% Tween 20 (TBST), prior to incubation with primary antibody. The membranes were then washed with TBST followed by incubation with an appropriate horseradish peroxidase-conjugated secondary antibody for 2 hr. Blots were treated with Clarity Western ECL kit, visualized on a ChemiDoc XRS+ (BioRad), and quantified using Image Lab Software (RRID: SCR_014210) according to standard procedures with experimental bands normalized to a control protein. Please refer to supplementary information for product numbers and Research Resource Identifiers of antibodies used in this study (*Supplementary file 1*).

## Primary cardiomyocyte isolation

Cardiomyocytes and cardiac fibroblasts at specified time points were isolated from FVB/NJ wild-type mice as previously reported (*Giudice et al., 2014*). Briefly, neonatal cardiomyocytes were isolated with a neonatal rat/mouse cardiomyocyte isolation kit (Cellutron Life Tech Highland Park, NJ, USA; nc-603[1]) using manufacturer's instructions. Cells from 24 to 36 hearts were pooled, pre-plated for 2 hr on an uncoated dish to separate fibroblasts from cardiomyocytes. The supernatant containing the cardiomyocytes was removed, plated on SureCoat (Cellutron Life Tech; sc-9035) coated plates, and incubated at 37°C in a humidified incubator with 5% CO2. After 12 hr, media was changed to NW (Cellutron Life Tech; m-8032) and cultured until use. The purity of cultures was determined by Western blot as well as immunofluorescence staining with anti-Desmin (Abcam, ab15200) and anti-Vimentin (Abcam, 11256) antibodies. The plate containing fibroblasts was washed with PBS three times before extracting protein/RNA. Adult cardiomyocytes were isolated from 2-month-old FVB mice using a Adumyt Cardiomyocyte Isolation Kit (Cellutron Life Technology, ac-7034) according to the manufacturer's instructions. Mice were treated with anticoagulant (500 U heparin i.p.) 30 min prior to heart extractions. Langendorf perfusion was carried out at 37°C. Cardiomyocytes and fibroblasts were separated using the same pre-plating method as described above.

## siRNA experiments and isoproterenol or T3 treatment

To evaluate the role of PABPC1 proteins in cardiac hypertrophy, isolated neonatal cardiomyocytes were cultured in serum containing NS media (Cellutron Life Tech) for 12 hr. After 12 hr, the cells were washed twice with pre-warmed Serum-free media (NW, Cellutron Life Tech) before transfecting with the siRNA against endogenous *Pabpc1* ORF or 3'- UTR using Lipofectamine RNAiMAX Transfection Reagent (Thermo Fisher). siRNA against *Luciferase* was used as control. After 12 hr, cells were washed with NW media and treated with 20 nM isoproterenol (Iso) or 20 ng of triiodothyronine (T3) and cultured for additional 36 hr.

## Cell culture and RNA extraction

C2C12 cells (ATCC, CRL-1772; RROD: CVCL_0188) were cultured in DMEM (Dulbecco's Modified Eagle's Medium) supplemented with 10% FBS, 2 mM glutamine, 100 units/mL penicillin and 100 µg/mL streptomycin and were maintained at 37°C in 5% CO2 as previously descried (*Singh et al., 2014*). The C2C12 cells were seeded in six-well plates. After 12–16 hr, when cell confluence reached approximately 100% the differentiation of C2C12 myoblasts into myotubes was induced by the addition of differentiation medium (DMEM containing 2.5% horse serum). Starting at the beginning of

differentiation, the C2C12 cells were cultured in six-well plates and harvested for RNA extraction at 0, 1, 2, 3, and 4 days after differentiation. Total RNA samples were extracted using TRIZOL (Invitrogen) according to the manufacturer's instructions. Upon DNAse treatment (Promega), RNAs (~2.5 µg) were reverse transcribed using random hexamer primers and Maxima Reverse Transcriptase kit (Thermo Scientific). The cDNA was diluted to 25 ng/µL with nuclease-free water and used for downstream qPCR assays.

## Luciferase assay

The 3' UTR of *Pabpc1* was cloned at the Not1 and Xho1 site of the psiCheck2 plasmid (downstream of RLuc). The 5' UTR of *Pabpc1* was cloned at Nhe 1 site (Upstream of RLuc). The double clone included both 5'UTR and 3'UTR and were cloned in their respective sites (Upstream and downstream of RLuc). C2C12 cells were plated on six-well plates at approximately 90–100% confluency and reverse transfections were performed using Mirus-Trans 20 reagent according to slightly modified manufacturer's protocol. Briefly, we transfected 3–5 µg of the respective plasmid at very high density of cells. The media was replaced after 24 hr of transfection with fresh low-serum differentiation media and cultured for 5 days before determination of luciferase activities using the Dual-Luciferase system. The Firefly and Renilla luciferase activities were measured at 24 hr after transient transfection for myoblast and 5 days after changing to low-serum differentiation media for myotube using the Dual-Glo Luciferase assay system (Promega) according to the manufacturer's instructions. Each plasmid was tested in three independent experiments. Luciferase activity was normalized using the Firefly luciferase activity levels and expressed as relative luciferase units (RLU) to reflect the influence of 5' UTR and 3'UTR activity on translation of Renilla luciferase.

## In vitro binding assays

The PABPC1-binding domain of eIF4G1 (residues 161–225) was cloned into the pET41a vector. The ORF of PABPC1 and PABPC1$_{mRRM2}$ were cloned into the pGX-2T vector to be expressed as N-terminal 6X His-tagged and C-terminal GST tagged fusion proteins respectively in *E. coli* BL21 (strain α-DE3). Cells expressing GST-fusion proteins were lysed in PBS (19 mM Na2HPO4, 0.9 mM KH2PO4, 2.5 mM KCl, 140 mM NaCl [pH 7.4]) and the proteins were purified by affinity chromatography on glutathione-magnetic beads (Thermo Fisher, 88821). Cells expressing His-tagged proteins were lysed in lysis buffer (20 mM Tris-Cl, 300 mM NaCl, 1% NP40. pH-7.4), and the proteins were purified by affinity chromatography on Ni$^{2+}$-NTA (Qiagen, 30230). The beads were washed using lysis buffer supplemented with 20 mM imidazole then eluted in 300 mM Imidazole buffer (20 mM Tris-Cl, 300 mM Imidazole, 10% Glycerol, 300 mM NaCl, pH 7.4). Equimolar amounts of His-eIF4GI$_{(161-225\ a.a)}$ and GST-PABPC1 or GST-PABPC1$_{mRRM2}$ were mixed and incubated with or without RNase A in 1XPBS for 3 hr at 4°C. Protein complexes were separated using Glutathione-Magnetic beads by incubating for 2 hr at 4°C. For control, purified GST protein was incubated with His-eIF4GI$_{(161-225\ a.a)}$. Magnetic beads with the protein complex were washed with ice cold 1X PBS and directly boiled in 2X laemmli buffer before separating on 10% SDS-PAGE gel. The protein complexes were visualized using Coomassie stain.

## Poly(A)-RNA-binding assay

Poly(A)25 RNA (IDT) was radiolabeled at the 5' end with 50 mCi $\gamma-32$P-ATP using T4 polynucleotide kinase (NEB). For filter binding assays, 1 fmol of $^{32}$P-poly(A)25 was incubated with purified GST, GST-PABPC1, or GST-PABPC1mRRM2 in a final volume of 100 µL (Tris-HCl buffer (pH 8.0),70 mM KCl, 10% glycerol, 0.05% IGEPAL, 1 mM DTT, 100 ng/mL BSA supplemented with 0.4 unit/mL RNasin). The reaction was incubated at 40°C for 3 hr on a 96-well dot-blot (Minifold I; Schleicher & Schuell Cat.10447900). A Hybond N+ nylon membrane was placed beneath a Amersham Hybond ECL Nitrocellulose Membrane and pre-equilibrated with 1X TBE. These membranes, used to capture Protein-RNA probe complexes and unbound RNA probes respectively, were further sandwiched in dot blot assembly. The incubated mixture was drawn through slowly by vacuum. Following incubation, membranes were separated, air dried, and measured by autoradiography for 6 hr. After 6 hr, the dots from membranes were cut out and corresponding radiation was quantified using a scintillation counter.

## Co-immunoprecipitation assay

C2C12 cells were expanded according to previously described methods. In brief, cells were thawed from a low passage number and expanded by splitting before 40% confluency. At 95–100% confluency, cells were infected with AdPABPC1, AdPABPC1$_{mRRM2}$, or AdGFP virus at $5 \times 10^9$ o.p.u and incubated for 4 days. After 4 days, the cells were washed with PBS and lysed on ice in lysis buffer (50 mM HEPES [pH 7.5], 150 mM NaCl, 0.5 mM EDTA, 10% glycerol, 1% Triton X-100, 5 mM dithiothreitol) supplemented with complete EDTA free protease inhibitor (Sigma-Aldrich, P0044) and phosphatase inhibitor (Thermo Scientific, 88666). Of cell lysate, 1–5 mg was incubated with 50% slurry of 200 µL of anti-Flag magnetic beads (Sigma-Aldrich, M8823) in each IP reaction and incubated at 4°C for 6 hr. After washing, the proteins were separated by 8% SDS-PAGE, transferred to PVDF membranes, and detected with anti-eIF4G1 antibody (#2617, Cell Signaling Technology).

## Adenovirus design and use

Adenovirus was produced using the methodology as previously described (Bhate et al., 2015). Briefly, ORF encoding FLAG-tagged PABPC1 and PABPC1mRRM2 were cloned into the p-Adeno-X-ZsGreen1 vector (Clonetech, 632267) using the In-Fusion kit (Clonetech, 639646) as per the manufacturer's instructions. High-titer adenoviruses were generated by transfecting Ad-293 cells (~70% confluent) in T-25 flasks with linearized recombinant adenoviral plasmid using Mirus TransIT-2020 reagent. Virus was harvested once a cytopathic effect was observed. Next, two viral amplification steps were performed and the viral particles were purified using a CsCl gradient according to the Adeno-X Adenoviral System 3 user manual. After purification of viral particles, the titer was determined by ultraviolet spectrophotometry at 260 nm. Neonatal cardiomyocytes cultured in NW media were transfected with siRNA target the 3'UTR of endogenous *Pabpc1*. After 6 hr, the cardiomyocytes were infected with $5 \times 10^9$ o.p.u. of Ad-PABPC1, Ad-PABPC1$_{mRRM2}$, or Ad-GFP adenovirus. Cells were washed and treated with 20 nM Isoproterenol following a 12 hr incubation with virus. After 36 hr of Isoproterenol treatment, cells were collected to harvest protein and RNA for further analysis.

## Protein synthesis rate, Click chemistry

Protein synthesis was measured using the L-homopropargylglycine (HPG) Click-iT (ThermoFisher, C10428) metabolic labeling reagents according to the manufacturer's protocol. Briefly, cultured neonatal cardiomyocytes were washed twice with warmed PBS and incubated in methionine-free DMEM supplemented with Iso or T3 for 1 hr. The medium was replaced with methionine-free DMEM to which 50 µM of the methionine analog HPG was added. Cells were again treated with same concentration of Iso or T3 and incubated for 60 min for incorporation of the AHA into nascent proteins with or without Iso/T3. After incubation, the dishes were rinsed twice. Newly synthesized proteins labeled with Click-iT HPG were conjugated with the carboxytetramethylrhodamine alkyne (TAMARA) using the Click-iT TAMRA Protein Analysis Kit (Cat. no. C33370). Protein samples were separated on 10% SDS-PAGE and was visualized using 532 nm excitation. The gel was subsequently stained with Coomassie blue for normalization.

## RNA-FISH

To determine the cellular location of *Pabpc1* mRNA, we performed RNA-FISH on undifferentiated and differentiated C2C12 cells using the Stellaris RNA-FISH kit (Biosearch Technologies). In brief, 48 fluorescently labeled oligonucleotide probes targeting *Pabpc1* mRNA were designed using the custom Stellaris Probe Designer (*Supplementary file 1*). Undifferentiated and differentiated C2C12 cells were fixed in a 3.7% formaldehyde buffer in 1X PBS. The cells were then permeabilized using 70% ethanol for 48 hr at 4°C. All subsequent steps were performed in the dark to minimize loss of fluorescent signal. Hybridization with the probes was performed by washing the cells with Stellaris Wash Buffer A before incubating overnight at 37°C with hybridization buffer containing the probe set or hybridization buffer only as a negative control. Cells were washed for 1 hr with Wash Buffer A before incubating with NucBlue (ThermoFisher Scientific) nuclear stain for 5 min. After incubation with DAPI, cells were washed in Wash Buffer B for 5 min and mounted with CC/Mount (Sigma-Aldrich). Images were obtained using a Zeiss LSM 700.

## RNAse H digestion and northern blotting

To assess the *Pabpc1* poly(A) tail lengths by northern blot, we mixed total RNA with 0.5 µM DNA oligonucleotides that hybridize to PABPC1 or GAPDH in 15 µL. Where indicated, oligo-dT40 was also included at 0.5 µM. After incubation at 65° for 5 min and chilling on ice, the following components were added to the reaction in 30 µL total volume: 1X RNase H buffer (Promega), 10 mM DTT, 15 ng/µL poly(A) (Sigma), 20 U RNasin (Promega), and 1 U RNase H. The reaction proceeded at 37° for 2 hr and was stopped by addition of 270 µL of G-50 buffer (0.25% SDS, 0.3 M NaOAc, 20 mM Tris pH 6.8, and 2 mM EDTA). RNA was isolated by standard phenol:chloroform:isoamyl alcohol (25:24:1) extraction followed by ethanol precipitation. Northern blots were performed as previously described (*Bresson and Conrad, 2013*). RNA probes were generated by incorporation of $^{32}$P-UTP into in vitro transcribed RNAs using T7 templates generated by PCR.

## Poly(A) tail fractionation

This protocol was based on one previously reported (*Kojima et al., 2015*). Herein, a 1X SSC solution contains 150 mM NaCl and 15 mM sodium citrate at pH 7.0. Briefly, 100 µL of GTC buffer (4M guanidine thiocyanate, 25 mM sodium citrate, pH 7.1) was mixed with ~5–15 µg RNA, 2 µL $\beta$-mercaptoethanol, and 3.75 µL of 50 µM biotinylated oligo dT (Promega) in a final volume of ~115 uL. To this mixture, 209 µL of dilution buffer (3X SSC, 5 mM TrisHCl pH 7.5, 0.5 mM EDTA, 0.125% SDS, 5% $\beta$-mercaptoethanol) was added. The samples were heated to 70°C for 5 min and centrifuged at 12,000 x g at room temperature. The supernatant was then mixed with MagneSphere Streptavidin paramagnetic particles (Promega; 150 µL of manufacturer's slurry) that had been washed three times with 0.5X SSC and Igepal-CA-630 was added to 0.1%. The RNA and dT and bead solution was allowed to bind for 15 min at 25°C while rotating and subsequently washed three times with 0.5X SSC containing 0.1% Igepal-CA-630. Elutions were performed by incubation of the beads in 400 µL SSC at the indicated concentration plus 0.1% Igepal-CA-630 for 5 min at 25°C. RNA was isolated by standard phenol:chloroform:isoamyl alcohol (25:24:1) extraction followed by ethanol precipitation. RNase T1 treatment and poly(A) tail northern blots were performed as previously described[10]. After standardization the elution conditions were then used on 2.5 µg of total RNA from heart tissue. In order to elute short poly(A) tailed mRNA [Poly(A)<30], we used elution buffer containing 0.075X SSC while for eluting long polyA mRNA [poly(A)>30] we used nuclease-free water. Eluted sample was purified and used for cDNA synthesis and downstream qPCR analyses.

## Statistics

All quantitative experiments (for example, qPCR, Western blots, cell areas and counts) have at least three independent biological repeats. Differences between groups were examined for statistical significance using Student's *t*-test (for two groups), or one-way ANOVA plus Dunnett's post-hoc test (for more than two groups) using the GraphPad Prism 6 Software (RRID: SCR_002798). Results were expressed as mean ± s.d., unless otherwise specified. *$p < 0.05$, **$p < 0.005$, ***$p < 0.001$ were considered statistically significant.

## Acknowledgements

The authors thank the members of the Kalsotra laboratory, Dr. Sayee Anakk, Dr. Stephanie Ceman, Dr. Robert L. Switzer, Dr. Manish Jaiswal and Dr. Ravi Singh for their valuable discussions and comments on the manuscript. This research was supported through NIH grants (R01HL126845) to AK, (R01AI081710) to NKC, and (R01GM111816) to JY. JS was supported partly by a National Institute of General Medical Sciences (NIGMS)–NIH Chemistry–Biology Interface Training Grant (5T32-GM070421). All authors declare no competing financial interests.

## Additional information

### Funding

| Funder | Grant reference number | Author |
| --- | --- | --- |
| National Heart, Lung, and | R01HL126845 | Auinash Kalsotra |

| | | Blood Institute |
| --- | --- | --- |
| National Institute of Allergy and Infectious Diseases | R01AI081710 | Nicholas K Conrad |
| National Institute of General Medical Sciences | R01GM111816 | Jing Yang |
| National Institute of General Medical Sciences | 5T32-GM070421 | Joseph Seimetz |

The funders had no role in study design, data collection and interpretation, or the decision to submit the work for publication.

### Author contributions

SC, Conceptualization, Data curation, Formal analysis, Investigation, Writing—original draft; JS, Data curation, Formal analysis, Investigation, Methodology, Writing—original draft, Writing—review and editing; RE, Data curation, Formal analysis; JY, Resources, Investigation; SMB, Data curation, Formal analysis, Investigation; MDL, Resources, Formal analysis, Investigation; GP, Resources; NKC, Conceptualization, Formal analysis, Supervision, Investigation; AK, Conceptualization, Formal analysis, Supervision, Funding acquisition, Investigation, Project administration, Writing—review and editing

### Author ORCIDs

Auinash Kalsotra, http://orcid.org/0000-0002-1011-0006

### Ethics

Animal experimentation: We followed the NIH guidelines for use and care of laboratory animals, and all experimental protocols were approved by IACUC (#A3118-01) at the University of Illinois, Urbana-Champaign.

## Additional files

### Supplementary files

• Supplementary file 1. This document contains detailed information on tools used in the study. Sheet 1 contains the forward and reverse sequences of all primers organized by RT-qPCR, genotyping of mouse lines, cloning for plasmid constructs, RNAse H cleavage, and generation of northern blot probes. Sheet 2 contains antigen, manufacturer, product number, and RRID information for all antibodies used. Sheet 3 contains the probe sequences for *Pabpc1* RNA-FISH.

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
