## [Decision Letter]

Thank you for submitting your article "Poly(A) tail length regulates PABPC1 expression and tunes translation in the heart" for consideration by *eLife*. Your article has been favorably evaluated by Fiona Watt (Senior Editor) and three reviewers, one of whom, Nahum Sonenberg (Reviewer #1), is a member of our Board of Reviewing Editors.

The reviewers have discussed the reviews with one another and the Reviewing Editor has drafted this decision to help you prepare a revised submission.

Summary:

The authors show that PABPC1 expression in the heart (cardiomyocytes) is regulated to support active translation during development and down-regulated to match the significantly reduced rate of translation observed in adult heart. The down-regulation of PABPC1 expression appears to be related to altered poly(A) status of the *Pabpc1* mRNA. The authors further reveal that PABPC1 expression is reestablished during exercise-induced hypertrophy to support the translation required for cardiomyocyte hypertrophy and that, in a model of chronic adult PABPC1 expression, that this adult PABPC1 expression does not appear to have obviously deleterious consequences.

The paper is of great interest to the readers in the fields of translational control and cardiology. In general, this is a high-quality study that reveals a previously unknown physiological role for PABPC1 and is therefore of significant interest. The data is generally of high quality and the interpretation is reasonable.

Essential revisions:

1) The authors state that they uncovered a poly(A)-based regulatory mechanism. There is a change in poly(A) tail length of *Pabpc1* mRNA, but data supporting a mechanistic role for this is missing. The questions that need to be answered are: What is the poly(A) tail status of the reporter mRNAs in C2C12 cells? What is the poly(A) status of *Pabpc1* mRNA in the different fractions across the gradient in E18 and adult heart? Is there a correlation between polysome association and poly(A) tail length, does poly(A) tail length drive polysome association or does it protect/enhance deadenylation? It needs to be clear whether changes in poly(A) tail length are a cause, effect or bystander. The experiment in RRL is not informative and should be deleted.

2) Polysome profiles should be provided for the PABPC1 overexpression in mouse heart experiment to establish that this does indeed result in a change in translational utilization of mRNAs as posited.

3) Delete Figure 1—figure supplement 1 – [or provide validation for the antibody]; Figure 2—figure supplement 3 – *Pabpc1* mRNA levels should not be plotted on a log scale – which masks any changes between fetal and adult heart. For Figure 1 -please provide single channel images in main/supplemental figures; clarify how protein levels were quantified (e.g. in WB- digital acquisition system), some stats are missing in places e.g. Figure 1, Figure 2—figure supplement 3.

4) Figure 2: If the model of poly(A) tail length regulation is correct, then one would expect that polysome-associated *Pabpc1* mRNA would increase under stress conditions. The addition of these experimental data would bolster the overall conclusion of the study.

5) Results and Discussion, seventh paragraph, Figure 3 and Figure S3—figure supplement 2A. Although PABPC1 plays an important role in mRNA stability in addition to translation, there are increased levels of *Acta1, Myh7* and *Anp* mRNAs in *Pabpc1*-knockown (KD) cells. Why are these mRNAs so stable in the absence of PABPC1?

6) Results and Discussion, in the conclusion, the authors need to address the question as to why is the *Pabpc1* mRNA harboring a short poly(A) tail stable in mature cardiomyocytes in the absence of PABPC1?

---

## [Author Response]

*Essential revisions:*

*1) The authors state that they uncovered a poly(A)-based regulatory mechanism. There is a change in poly(A) tail length of PABPC1 mRNA, but data supporting a mechanistic role for this is missing. The questions that need to be answered are: What is the poly(A) tail status of the reporter mRNAs in C2C12 cells?*

We appreciate the reviewer’s suggestion to measure poly(A) tail lengths of *Pabpc1* reporter mRNAs. This is an interesting experiment, but somewhat artificial because our reporters contain strong heterologous polyadenylation elements as part of the plasmid backbone. Instead, we measured the poly(A) tail status of endogenous *Pabpc1* mRNAs in C2C12 cells. The poly(A) tail length data obtained were consistent with reduced PABPC1 protein levels seen during myoblast-to-myotube differentiation (Figure 2—figure supplement 2) such that *Pabpc1* mRNAs were significantly deadenylated in myotubes when compared to myoblasts. These new data are added to the revised Figure 2.

*What is the poly(A) status of PABPC1 mRNA in the different fractions across the gradient in E18 and adult heart? Is there a correlation between polysome association and poly(A) tail length, does poly(A) tail length drive polysome association or does it protect/enhance deadenylation? It needs to be clear whether changes in poly(A) tail length are a cause, effect or bystander.*

We have addressed this concern of the reviewer to provide further evidence for the correlation of *Pabpc1* poly(A) tail length and polysome association. An additional experiment measuring *Pabpc1* mRNA levels in short and long poly(A) tail fractions from polysome gradients has been added. In the neonatal heart, *Pabpc1* has long poly(A) tail length and associates exclusively with polysomes, whereas in the adult heart most *Pabpc1* mRNAs are short-tailed and associate with monosome and RNP fractions (Figure 2). Poly(A) tail length of *Gapdh* mRNAs remain unchanged and primarily associate with polysomes at both developmental stages (Figure 2—figure supplement 3). Thus, these results point towards a causal effect of poly(A) shortening in suppressing *Pabpc1* translation in the adult heart.

*The experiment in RRL is not informative and should be deleted.*

As requested, we have removed the RRL in vitro translation data from the revised Figure 2.

*2) Polysome profiles should be provided for the PABPC1 overexpression in mouse heart experiment to establish that this does indeed result in a change in translational utilization of mRNAs as posited.*

We appreciate the reviewer’s suggestion to demonstrate the effect of PABPC1 re-expression on translation. However, we feel that polysome profiles are, by nature, qualitative and believe that our in vivo experiment utilizing the SUnSET method in PABPC1 re-expressing mice (Figure 5) is not only quantitative, but also more informative regarding the impact of PABPC1 on global translation in the heart. The SUnSET assay has become widely accepted in the field to measure protein synthesis and has been previously demonstrated to match polysome traces (Genes Dev. (2016) 30:1-6; Nat Commun (2016) 11776; FASEB J (2017) 30; 2: 798-812).

*3) Delete Figure 1—figure supplement 1 – [or provide validation for the antibody].*

Anti-PABPC1 antibody (Abcam ab21060) used in the Human protein atlas database (see http://www.proteinatlas.org/ENSG00000070756-PABPC1/antibody) has been previously validated to specifically cross-react with human PABPC1. siRNA-based validation of this antibody is available in J Virol. 2013 Jan;87(1):243-56.

*Figure 2—figure supplement 3 – PABPC1 mRNA levels should not be plotted on a log scale – which masks any changes between fetal and adult heart.*

Thank you for raising this point. We have revised the Figure 2—figure supplement 2 by plotting the PABPC1 mRNA and protein level data on a normal scale in the graph.

For Figure 1 -please provide single channel images in main/supplemental figures.

As requested, we have now provided the single channel images in the Figure 1—figure supplement 2.

Clarify how protein levels were quantified (e.g. in WB- digital acquisition system).

We have expanded the description of methods used to quantify the protein levels. Briefly, the blots were treated with an ECL substrate, visualized on a ChemiDoc digital acquisition system, and quantified using Image Lab Software.

*Some stats are missing in places e.g. Figure 1, Figure 2—figure supplement 3.*

We thank the reviewers for pointing out this oversight. We have now added the statistics, *p* values and *n* numbers to Figure 1 and Figure 2—figure supplement 2. The western blot data in Figure 1, is derived from a single pool of fetal and adult human heart protein samples and this information is provided under the Methods section.

*4) Figure 2: If the model of poly(A) tail length regulation is correct, then one would expect that polysome-associated Pabpc1 mRNA would increase under stress conditions. The addition of these experimental data would bolster the overall conclusion of the study.*

We understand the reviewer’s curiosity about the increase in polysome association of *Pabpc1* mRNA under stress conditions. These lines of investigations are logical extensions of our work, which we will be entertaining and testing in the future but they fall outside the scope of this current study.

*5) Results and Discussion, seventh paragraph, Figure 3 and Figure S3—figure supplement 2A. Although PABPC1 plays an important role in mRNA stability in addition to translation, there are increased levels of Acta1, Myh7 and Anp mRNAs in PabpC1-knockown (KD) cells. Why are these mRNAs so stable in the absence of PABPC1?*

We thank the reviewers for raising this important point. As has been previously reported, we believe that increased mRNA abundance of *Acta1, Myh7* and *Anp* in cardiomyocytes after Isoproterenol or T3 stimulation is due to increased transcription of these hypertrophy markers (J Clin Invest. (2008) 118(1): 124-132; Biochim Biophys Acta. (2013) 1832 (12): 2403-13). However, as the reviewer pointed out, it is interesting to note that the increased mRNA abundance of *Acta1, Myh7* and *Anp* is maintained even in PABPC1 depleted cells after Iso or T3 stimulation. These results imply that PABPC1 knockdown does not (i) interfere with the transcriptional regulatory circuits; and/or (ii) have a major impact on the mRNA stability of these genes. We have added this discussion to the revised manuscript.

*6) Results and Discussion, in the conclusion, the authors need to address the question as to why is the Pabpc1 mRNA harboring a short poly(A) tail stable in mature cardiomyocytes in the absence of PABPC1?*

Thank you for this comment. It is indeed interesting that in spite of harboring short poly(A) tails, *Pabpc1* mRNA is stable in mature cardiomyocytes. This stabilization could be imparted through binding of *trans*-acting factor(s), due to the presence of RNA structural element(s), or RNA modifications that may inhibit further recruitment/activity of deadenylases. We have mentioned these possibilities now in the Discussion section.